# Inositol 1,4,5-Trisphosphate Receptors in Human Disease: A Comprehensive Update

**DOI:** 10.3390/jcm9041096

**Published:** 2020-04-12

**Authors:** Jessica Gambardella, Angela Lombardi, Marco Bruno Morelli, John Ferrara, Gaetano Santulli

**Affiliations:** 1Department of Medicine, Einstein-Mount Sinai Diabetes Research Center (ES-DRC), Fleischer Institute for Diabetes and Metabolism, Albert Einstein College of Medicine, New York, NY 10461, USA; jessica.gambardella@einsteinmed.org (J.G.); angela.lombardi@einsteinmed.org (A.L.); marco.morelli@einstein.yu.edu (M.B.M.); j.ferraraeastchester@gmail.com (J.F.); 2International Translational Research and Medical Education Consortium (ITME), 80100 Naples, Italy; 3Department of Advanced Biomedical Sciences, “Federico II” University, 80131 Naples, Italy; 4Department of Microbiology and Immunology, Albert Einstein College of Medicine, New York, NY 10461, USA; 5Department of Molecular Pharmacology, Wilf Family Cardiovascular Research Institute, Albert Einstein College of Medicine, New York, NY 10461, USA

**Keywords:** Alzheimer, ataxia, autoimmune disease, cancer, cardiovascular disease, diabetes, GWAS, IP3 Receptors, ITPRs, mutations

## Abstract

Inositol 1,4,5-trisphosphate receptors (ITPRs) are intracellular calcium release channels located on the endoplasmic reticulum of virtually every cell. Herein, we are reporting an updated systematic summary of the current knowledge on the functional role of ITPRs in human disorders. Specifically, we are describing the involvement of its loss-of-function and gain-of-function mutations in the pathogenesis of neurological, immunological, cardiovascular, and neoplastic human disease. Recent results from genome-wide association studies are also discussed.

## 1. Introduction

Since their discovery in the 1970s, several studies have provided substantial evidence that inositol 1,4,5-trisphosphate receptors (ITPRs) play a pleiotropic role in the regulation of cellular functions. Indeed, their ability to regulate calcium handling poses ITPRs at the heart of molecular networks underlying cellular homeostasis: From proliferation, apoptosis, and differentiation to metabolism and neurotransmission.

ITPR was identified for the first time as a large membrane protein called P400 [1,2] that was able to regulate intracellular calcium spikes [1,2,3]. After protein purification and cDNA isolation, it became clear that P400 was a channel releasing calcium from the endoplasmic reticulum (ER) [4,5,6]. Later, ITPR was shown to be a rather peculiar channel, as two-second messengers are needed for its activation: IP3 and calcium [7,8,9,10,11,12]. 

Three isoforms of ITPR have been identified (ITPR13) in mammals, which, albeit produced by different genes, show 70% of homology in the primary protein sequence [13]. The similarity in amino acid sequence also reflects the resemblance in protein conformation and spatial organization. All three isoforms consist of five domains: Suppressor domain (SD), IP3 binding core domain (IBC), regulatory domain, transmembrane domain (TD), and C-terminus domain (CTD) [14,15,16,17]. These domains are organized in a complex tetrameric “mushroom-like” structure (Figure 1), with the stalk inserted in the ER membrane and the cap exposed to the cytosol [18]. The stalk is mainly represented by the transmembrane TM domain, with its six-helices forming the ion-conducting pore [16]. All the other domains are in the “cap”, exposed to the cytosol. This organization makes the IBC domain available to IP3 binding, and the regulatory domain to the many interactions and post-transcriptional modifications that regulate the receptor activity, including phosphorylation and oxidation [16]. 

Nevertheless, the information on the ITPR molecular organization is still not sufficient for a complete mechanistic definition of its structure–function relationship [10,19]. If the central calcium conducting pore is similar to other ion channels, as suggested by the 4.7 Å structure of ITPR [16], the spatial arrangement of the cytosolic C-terminus is quite unique for ITPR; in particular, these carboxyl tails have the ability to interact with the N-terminal domains of the near subunits, suggesting a mechanism of allosteric regulation dictated by intracellular signals [16]. The feature of ITPR of being prone to modulation by nearby signals gives an idea of the complexity of the ITPR-interactome. In other words, ITPRs have the structural complexity to participate in and regulate a dense network of cellular processes. ITPRs are differently expressed in human tissues, as reported in Table 1, obtained with data retrieved from the Human Protein Atlas [20]. The effects of ITPRs have been extensively studied in preclinical models [21,22,23,24,25,26,27,28,29]. Here, we offer an overview of the human pathologies where ITPR alterations have a clear causative role. Moreover, we summarize the information derived from innovative studies of the disease-genome profile association, which also suggests the potential, under-investigated role of ITPR in several human pathologies.

## 2. ITPRs and Neurological Disorders

The function of ITPR has been historically assessed in the neurological field. Indeed, the first identification of P400 protein occurred in Purkinje cells and the neurological signs were the first to be studied in mice [30]. The highest number of ITPR human mutations has been identified in neurological disorders, in particular affecting the isoform 1. Indeed, ITPR1 is the most abundant isoform in the brain, regulating important functions including memory and motor coordination [31].

### 2.1. Spinocerebellar Ataxia

Spinocerebellar ataxia (SCA) is a term referring to a group of hereditary ataxias characterized by degenerative alterations in the part of the brain related to the movement control (cerebellum) and sometimes in the spinal cord. Van de Leemput was the first to identify the deletion of a 5′ portion of ITPR1 in British and Australian families with type 15 SCA [32]. Thereafter, the deletion of exons 1-48 of ITPR1 was identified in other populations, demonstrating that the haploinsufficiency of ITPR1 is involved in SCA15-16 [33,34]. Missense mutations in the ITPR1 gene have been later associated with SCA15: P1059L and P1074L in a Japanese family, and V494I in an Australian family [35,36].

Another form of SCA, SCA29, characterized by an early-onset motor delay, hypotonia, and gait ataxia, is one of the forms more frequently associated with ITPR1 mutations [37]. The missense mutations V1553M and N602D, identified by Huang et al. [38], are among the first mutations observed; G2547A was identified as a de novo mutation but only in one case [39]. In a cohort study on a population of 21 patients with SCA29, Zambonin et al. identified six novel mutations in the ITPR1 gene [40]: Three mutations in the IP3 binding domain (R269G, K279E, K418ins), two mutations in the transmembrane domain (G2506R, I2550T), and one in the regulatory domain (T1386M); no specific genotype–phenotype correlations were observed, but the recurrence in affected subjects suggested the pathogenic role of these mutations.

SCA has been generally associated with a loss of function of ITPR1, however, Casey et al. recently identified a gain-of-function pathogenic mutation [41], detecting a R36C missense variant in three SCA29 affected members of the same family. The resultant ITPR1 mutant displayed a higher IP3 binding affinity than the wild type counterpart, converting the pattern of intracellular calcium release from transient to sigmoidal. This evidence supports the idea that the enhancement of calcium release can contribute to SCA29 pathogenesis. In addition to missense mutations of the ITPR1 gene, a splicing variant was also associated with SCA29: The c.1207-2A-T transition was identified in exon 14 of ITPR1 in four SCA29 patients and was not found in unaffected members of the same family [42].

Notably, all the mutations described above are autosomal dominant variants; however, a missense mutation in the ITPR1 gene was similarly associated with autosomal recessive SCA: In a family with congenital SCA history, the homozygous missense mutation L1787P was identified in all affected individuals, while the heterozygous carriers were asymptomatic [43]. The ability of this mutation to alter the receptor function is only predictive, but the concerned residue is highly conserved and the transition of leucine to proline can affect the protein stability with a high probability. Moreover, missense mutations (T267M, T594I, S277I, T267R) were observed in sporadic infantile-onset SCA [44,45], in congenital ataxias (R269W, R241K, A280D, E512K) [46], and in another subtype of ataxia, ataxic cerebral palsy (S1493D) [47]; other mutations have been reported [48,49,50,51] in molecularly unassigned SCA forms (V2541A, T2490M) and in rare forms of cerebellar hypoplasia (T2552P, I2550N).

Intriguingly, there are SCA variants not directly associated with ITPR1 gene mutations, but involving genes functionally close to ITPR1 and its signaling. A good example comes from SCA2 and SCA3, where the causative mutations are alterations in ataxin-2 and -3, respectively. In both cases, the mutant forms of ATXs are able to bind ITPR1 increasing the sensitivity of the channel for IP3 and enhancing channel gating [52,53]. Of interest, ITPR1-functional alterations by ATXs seem to have a pathogenic role, as they increase the apoptosis of Purkinje cells in animal models of ataxia [54]. This evidence supports the key role of calcium homeostasis regulation by ITPR1 in these neurological disorders even if the channel function is not directly altered.

### 2.2. Huntington’s Disease and Alzheimer’s Disease

To date, genetic mutations in the ITPR1 gene with a pathological relevance in human Huntington’s Disease (HD) and Alzheimer’s Disease (AD) have not been detected. However, both disorders are major examples of indirect involvement of ITPR1.

In HD, the causative mutation is the poliQ expansion of Huntingtin (Htt), although the cellular and molecular mechanisms of GABAergic neurons loss are not clearly understood [55]. Of note, the polyQ-Htt can bind ITPR1 with high affinity, sensitizing the receptor activity by IP3 [56,57]. Blocking the Htt–ITPR1 interaction in vivo was shown to regulate the abnormal calcium signaling in response to glutamate, protecting the neurons from death, and improving motor coordination [58], posing ITPR1 in a key position in HD pathophysiology and supporting the use of ITPR1-based therapies.

In AD, the pathogenic hypothesis of the beta-amyloid plaque has been extensively and historically investigated. However, in the last years, new and additional potential mechanisms have been suggested, including the dysregulation of calcium handling. In particular, ITPRs seem to have a key role in modulating calcium signals in AD [59]. Alterations in the ITPR function have been detected in cells derived from patients with AD already in 1994 [60,61]. Later, Ferreiro et al. demonstrated that antibody-aggregates are able to induce calcium release by ITPR in cortical neurons, leading to apoptosis, which was prevented by the ITPR inhibitor Xestopongin C [62].

### 2.3. Gillespie Syndrome

The Gillespie syndrome (GS) is a rare form of aniridia, cerebellar ataxia, and mental deficiency, described in 1965 by the American ophthalmologist Fredrick Gillespie [63]. Until 2016, its causative gene and mutations were unknown. Using a whole-exome sequencing approach, Gerber et al. identified several ITPR1 mutations in five GS-affected families [64]. In particular, the Authors detected truncating mutations in homozygous (Q1558*, R728*) or in composed heterozygous (G2102Valfs5/A2221Valfs23); the resultant truncated mutants were unable to generate a functional channel in a heterologous cell system. In the other two families, the Authors found one missense mutation (F2553L) and one deletion (K2563del) in the transmembrane domain. The latter, in addition to producing a dysfunctional channel, was able to exert a negative effect on the product derived from the wild type allele, with a dominant-negative action.

Later, next-generation sequencing approaches were used to study other GS families. The results evidenced novel missense mutations in the region of the calcium pore (N2543I), and in the regulatory domain (E2061G, E2061Q), further extending the ITPR1 mutations spectrum associated to GS [65,66].

### 2.4. Autism Spectrum Disorder

The Autism spectrum disorder (ASD) is a complex heterogeneous disorder with a poorly defined etiology and diagnosis criteria. Its high heritability, however, suggests a strong genetic component [67] and several genetic studies suggest that calcium homeostasis is a key determinant in its pathophysiology [68]. Recent studies demonstrate that IP_3_-mediated calcium signals are significantly depressed in fibroblasts isolated from patients with ASD, identifying ITPR as a functional target in this disease [69]. These data are consistent with another study done in patients with autism in which a genetic variant of the oxytocin receptor (implicated in the etiology of ASD), causes a decline in the IP3/calcium signaling pathway in vitro [70].

### 2.5. Amyotrophic Lateral Sclerosis

Amyotrophic lateral sclerosis (ALS) is a condition characterized by a progressive degeneration of motor neurons in the brain and spinal cord. ITPR1 and ITPR2 are the main isoforms expressed in motor neurons [71]. ITPR2 mRNA levels are elevated in peripheral blood samples of patients with ALS [72] and studies done in human cells suggest that the pharmacological inhibition of ITPR1 is a potential strategy to prevent motor neuron deterioration in ALS [73].

## 3. ITPRs in Autoimmune Disorders

ITPRs are important for exocrine fluid secretion including saliva, pancreatic juice, and tear secretion [74]. Interestingly, anti-ITPR antibodies have been detected in sera from patients with Sjogren’s syndrome (SS) [75], a chronic autoimmune disease involving lymphocytic infiltration and loss of secretory function in salivary and lacrimal glands [76]. A recent study demonstrates that the expression of ITPR2 and ITPR3 is significantly reduced in the salivary gland of SS patients, suggesting that deficits in ITPRs may underlie the secretory defect in SS [77].

Antibodies against ITPRs were also found in patients with rheumatoid arthritis and systemic lupus erythematosus, although the locations of the antigenic epitopes were different among the disease conditions [75].

## 4. ITPRs and Anhidrosis

Anhidrosis is the inability to sweat, which is responsible for heat tolerance; it is a rare disorder occurring even in the presence of morphologically normal eccrine glands, which are the main glands that respond to thermal stress with a high secretion rate. The whole-genome analysis of a family with anhidrosis and normal eccrine glands unveiled a novel missense mutation (G2498S) in ITPR type 2 [78]. This mutation occurs in the calcium pore-forming region. This association was corroborated by the observation of anhidrosis and hyperhidrosis in human pathologies linked to ITPR dysfunction; also, there was a marked reduction of sweat secretion in ITPR2^-/-^ animals. Interestingly, ITPR2 inhibitors have the potential to reduce sweat production in hyperhidrosis, suggesting that ITPR2 is a potential pharmacological target in the treatment of sweat secretion conditions [78].

## 5. ITPRs and Cancer

Calcium has a key role in proliferation, differentiation, and migration; therefore, it is not surprising that ITPR, one of the main regulators of calcium handling, is involved in neoplastic transformation and progression [79]. Neck squamous cell carcinoma (HNSCC) was one of the first diseases connected to ITPR [80]; a whole-exome sequencing analysis of HNSCC patients revealed missense mutations affecting the ITPR3 gene, R64H, and R149L, both in the regulatory domain of the receptor [80]. Importantly, ITPR3 gene mutations were detectable only in metastatic or in recurrent tumors, but not in the respective primary tumors. This finding strongly suggests a role for ITPR3 in the metastatic process and malignant transformation, very significant if we take into account that the major problem related to HNSCC is given by recurrent metastases, which occur in more than half of the patients.

An increased expression of ITPR3 was detected in clear renal cell carcinoma compared to the unaffected part of the kidney [81]; ITPR3 silencing affected tumor growth, in vitro as well as in vivo, providing a direct proof of the involvement of this receptor in the carcinogenesis. An increased expression of ITPR3 has been also observed in cholangiocarcinoma [82] and in colorectal cancer [83]; in both cases the expression of ITPR3 correlated with the degree of neoplasia severity [82,83].

Alterations of ITPR1 and ITPR2 have been associated with the Sézary syndrome, a T-cell lymphoma with an aggressive clinical course. The analysis of gene mutations in 15 patients with the neoplastic syndrome unveiled somatic point mutations in ITPR1 including A95T in the regulatory domain and S2454F in the trans-membrane domain, and mutation in ITPR2, such as S2508L, in the trans-membrane domain [84]. These discoveries have important implications in considering ITPRs as new therapeutic targets in cancer.

## 6. Potential Role of ITPRs in Human Disease: Evidence from GWAS

While an evident role of ITPRs has been recognized for several human pathologies by identifying specific mutations, a potential role of this channel in other human conditions has been suggested by genome whole association studies (GWASs). Eleftherohorinou et al. have shown that a defective second messenger signaling could be involved in the predisposition to rheumatoid arthritis [85]. Specifically, alterations in ITPRs were proposed to be responsible for calcium signaling deregulations in this disease [85]. As revealed by another GWAS, the involvement of ITPR3 in the release of the macrophage migration inhibitory factor (MIF) confirmed the role of this receptor in rheumatoid arthritis [86]. In the same study, ITPR was also associated with type 1 diabetes mellitus [86], reflecting the similarity of genetic perturbations and the comparable immunological dysfunctions underlying these diseases, further corroborated by genetic analyses identifying ITPR3 as an independent risk locus in Graves’ disease [87] and allergic disorders including asthma, allergic rhinitis, atopic dermatitis [88,89], and airflow obstruction [90]. The involvement of ITPR3 in diabetes has been also confirmed by the significant recurrence of single nucleotides polymorphisms (SNPs) in the ITPR3 gene in diabetic American women [91], as well as in a Swedish nationwide study [92]. The latter study reported that a variation at rs2296336 (a SNP within ITPR3) might influence the risk of developing diabetes through an effect on alternative splicing. Moreover, rs3748079, a SNP located in the promoter region of ITPR3, has been associated with several autoimmune diseases including systemic lupus erythematosus, rheumatoid arthritis, and Graves’ disease in a Japanese population [93], and the variant rs999943 of ITPR3 has been linked to obesity [94]. Equally important, ITPR1 has been associated in different GWASs with diabetic kidney disease [95] and obesity-related traits [96]. Of note, a recent GWAS in a Chinese population identified *ITPR2* as a susceptibility gene for the Kashin-Beck disease, a chronic osteochondropathy characterized by cartilage degeneration [97]; in this study, a significant association between the disease and nine SNPs of *ITPR2* was described. Interestingly, the regulatory role of ITPR2 in apoptosis is a possible contributor to the Kashin-Beck disease, since excessive chondrocyte apoptosis was found to be related to cartilage lesions in affected patients [98]. Moreover, a GWAS has revealed that the ITPR signaling pathway is genetically associated with epilepsy [99], and the anti-epileptic drug levetiracetum is known to act inhibiting the release of calcium by ITPRs, highlighting the relevance of enhanced ITPRs action in epilepsy [100].

Other association studies have underlined the role of ITPRs in the cardiovascular field. The association between gene expression and dilated cardiomyopathy (DCM) has been studied by assessing the presence of CpG sites in the proximity of gene-promoters, as an index of promoter methylation and consequent downregulation of transcription [101]; using this strategy, the CpG site “cg26395694” close to the ITPR1 locus (ENSG00000150995) has been shown to be significantly associated to DCM (*p*-value: 2.57E-02). More in general, ITPR3-mediated pathways have been also linked to ischemic heart disease [86] and coronary artery disease [102]. ITPR3 has been associated with the risk of developing coronary artery aneurism in Taiwanese children with Kawasaki disease [103], a multisystemic vasculitis that can result in coronary artery lesions and that had been linked to aberrant calcium signaling [104]. In a case-control study involving 93 Kawasaki disease patients and 680 healthy controls, the frequency of the rs2229634 T/T genotype was significantly higher in Kawasaki disease patients with coronary artery aneurism than in patients without coronary artery aneurism [103]. The key importance of ITPRs in cardiovascular medicine is confirmed by the crucial role of ITPRs in cardiogenesis [105,106,107]; ergo, it may be difficult to detect mutations causing severe heart defects by using genetic analyses of patient samples postnatally, especially if considering that ITPRs have been shown to be essential in very early embryogenesis and some mutations might cause lethality in utero [108,109].

An international GWAS identified ITPR1 between the novel loci associated with blood pressure in children and adolescents [110]. Finally, in the Hispanic population, ITPR1 was associated to the pathophysiology of childhood obesity [96], while ITPR3 was linked to body mass index variants conferring a high risk of extreme obesity [94].

The GWAS demonstrated an association of ITPR1–2 with different forms of cancer, especially breast cancer [111,112,113]. Other studies associated the expression level of ITPR3 with the aggressiveness of different types of tumors, including colorectal carcinoma, gastric cancers [114], and head and neck squamous cell carcinoma [80]. *ITPR3* variants were also found to be implied in cervical squamous cell carcinoma [115]. Interestingly, ITPR3 appears also to actively participate in cell death in several tissues and its increased activity was demonstrated to induce apoptosis in T lymphocytes [116,117]. These findings indicate that compounds aimed at controlling the ITPR activity may be useful as a therapeutic approach for modulating immune responses in cancer.

## 7. Conclusions

In this systematic review, we illustrated the association of ITPRs mutations with human disorders. The mutations of ITPRs reported in humans are summarized in Table 2 and represented in Figure 1. Throughout the analysis of current literature, the involvement of ITPRs in human disease appears to be under-investigated.

The currently known contribution of the receptor to the pathogenesis of human disease is only the top of the iceberg. The information about causative genetic alterations affecting ITPRs mainly come from the neurology-related fields, cancer fields, or rare disease field, where the genetic analysis is a more common approach included in diagnostic procedures. However, in several studies of large-scale genome analysis, ITPRs recurrently emerge as a susceptibility gene for several pathological conditions. This evidence confirms that only little is known about this channel, particularly in cardiac and vascular homeostasis or metabolism. The recent findings of the physical link between ER and mitochondria, mediated by a protein complex including ITPR, suggest a potential role of the receptor in the regulation of calcium-dependent mitochondrial metabolism [118,119,120,121,122,123,124,125,126,127,128,129,130]. The ability of ITPR to indirectly regulate mitochondrial energetic metabolism could have a significant impact on the health and homeostasis of the tissues strongly dependent on mitochondrial energetic production, such as cardiac and skeletal muscle. However, this aspect needs to be further explored.

The underestimated pathophysiological role of ITPR might also depend on the fact that the cellular context strongly affects the impact of ITPR alterations on calcium handling and the relative cell fate. A good example comes from a study in neuronal cells about the P1059L affecting the regulatory domain of ITPR1; this mutation increases the affinity of ITPR1 to IP3, altering the functional output of Purkinje cells, however, no differences were detected in calcium signaling between the wild type and the same mutant in B-cells [131]. The regulatory domain of ITPR is the target of several molecular partners whose expression and activity profile are different among the different cellular contexts. Therefore, the regulation around the P1059 residue of ITPR could be different—and/or of different impact—in Purkinje cells compared to other cells, such as B-cells. Nevertheless, the experiments performed in stable cell lines can alter the impact of ITPR alterations, as several adaptive pathways could affect the expression of ITPR regulatory proteins minimizing the effects of mutations. These observations encourage future studies on ITPR in the appropriate native cellular context, both in physiological and pathological conditions.

## Figures and Tables

**Figure 1 jcm-09-01096-f001:**
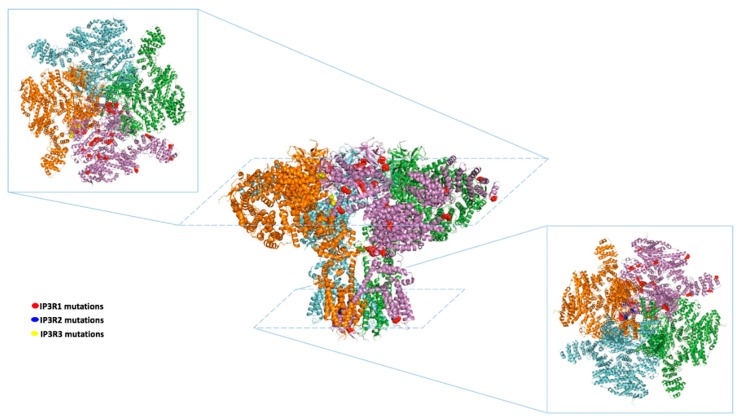
Representative structure of inositol 1,4,5-trisphosphate receptors (ITPRs) (1-3) showing disease-related mutations. In the middle, representative “mushroom-like structure” of ITPRs. For clarity, only the crystal of human isoform 3 is shown. Top left corner: View from the top; bottom right corner: View from the bottom. The residues in red, blue, and yellow indicate the mutations in ITPR1, 2, and 3, respectively, that have been hitherto reported in humans.

**Table 1 jcm-09-01096-t001:** Protein expression levels of IP3Rs in different human tissues and organs.

*Tissue*	IP3R1	IP3R2	IP3R3
*Cerebral cortex*	XX	XX	X
*Cerebellum*	XX	X	XXX
*Hippocampus*	XX		
*Caudate*	XX	X	
*Thyroid gland*		X	X
*Parathyroid gland*		XXX	
*Adrenal gland*		XX	X
*Nasopharynx*		X	XX
*Bronchus*		XX	XX
*Lung*	X	X	XX
*Oral mucosa*		X	XX
*Salivary gland*		XX	
*Esophagus*		X	XX
*Stomach*	X	XX	XX
*Duodenum*		XX	XX
*Small intestine*		XX	XXX
*Colon*		XX	XX
*Rectum*		XX	XX
*Liver*		XX	XX
*Gallbladder*		XX	X
*Pancreas*		XX	X
*Kidney*	X	XXX	X
*Urinary bladder*		X	XX
*Testis*	X	XX	XXX
*Epididymis*	X	XX	X
*Seminal vesicle*	X	X	X
*Prostate*	X		X
*Vagina*			XX
*Ovary*		X	
*Fallopian tube*		XX	X
*Endometrium*		XX	XXX
*Cervix, uterine*		X	XX
*Placenta*		X	X
*Breast*	X	XXX	XX
*Heart*	X	XX	
*Smooth muscle*		XX	
*Skeletal muscle*		XX	
*Soft tissue*			
*Adipose tissue*		XX	
*Skin*		XX	XX
*Appendix*		XX	XX
*Spleen*	X		
*Lymph node*	X		X
*Tonsil*	X	X	XXX
*Bone marrow*		X	

X: Low, XX: Medium, XXX: High protein expression level.

**Table 2 jcm-09-01096-t002:** Spectrum of IP3Rs mutations identified in humans.

Mutation	IP3R Isoform	Effect on Protein	Disease	Reference
*5′ deletion*	IP3R1	Downregulation	SCA15	[32]
*1-48 exons deletion*	IP3R1	Downregulation	SCA15-16	[33,34]
*P1059L*	IP3R1	Missense (ND)	SCA15	[35]
*P1074L*	IP3R1	Missense (ND)	SCA15	[35]
*V494I*	IP3R1	Missense (ND)	SCA15	[36]
*V1553M*	IP3R1	Missense (ND)	SCA29	[38]
*N602D*	IP3R1	Missense (ND)	SCA29	[38]
*G2547A*	IP3R1	Missense (ND)	SCA29	[39]
*R269G*	IP3R1	Missense (ND)	SCA29	[40]
*K279E*	IP3R1	Missense (ND)	SCA29	[40]
*G2506R*	IP3R1	Missense (ND)	SCA29	[40]
*I2550T*	IP3R1	Missense (ND)	SCA29	[40]
*T1386M*	IP3R1	Missense (ND)	SCA29	[40]
*R36C*	IP3R1	Gain-of-function Increase of IP3 binding affinity	SCA29	[41]
*c.1207-2A-T*	IP3R1	Splicing variant	SCA29	[42]
*L1787P*	IP3R1	Protein-instability*	Autosomal-recessive SCA	[43]
*T267M*	IP3R1	Missense (ND)	Sporadic infantile-onset-SCA	[44,45]
*T594I*	IP3R1	Missense (ND)	Sporadic infantile-onset-SCA	[44,45]
*S277I*	IP3R1	Missense (ND)	Sporadic infantile-onset-SCA	[44,45]
*T267R*	IP3R1	Missense (ND)	Sporadic infantile-onset-SCA	[44,45]
*R269W*	IP3R1	Missense (ND)	Congenital-ataxias	[46]
*R241K*	IP3R1	Missense (ND)	Congenital-ataxias	[46]
*A280D*	IP3R1	Missense (ND)	Congenital-ataxias	[46]
*E512K*	IP3R1	Missense (ND)	Congenital-ataxias	[46]
*S1493D*	IP3R1	Missense (ND)	Ataxic-cerebral-palsy	[47]
*V2541A*	IP3R1	Missense (ND)	Molecular-unassigned SCA	[48]
*T2490M*	IP3R1	Missense (ND)	Molecular-unassigned SCA	[48]
*T2552P*	IP3R1	Missense (ND)	Cerebellar-hypoplasia	[50]
*I2550N*	IP3R1	Missense (ND)	Cerebellar-hypoplasia	[51]
*Q1558*	IP3R1	Truncating-protein, no functional channel	Gillespie syndrome	[64]
*R728*	IP3R1	Truncating-protein, no functional channel	Gillespie syndrome	[64]
*F2553L*	IP3R1	Missense (ND)	Gillespie syndrome	[64]
*K2563 deletion*	IP3R1	Dysfunctional channel with dominant negative action	Gillespie syndrome	[64]
*N2543I*	IP3R1	Missense (ND)	Gillespie syndrome	[65]
*E2061G*	IP3R1	Missense (ND)	Gillespie syndrome	[66]
*E2061Q*	IP3R1	Missense (ND)	Gillespie syndrome	[66]
*A95T*	IP3R1	Missense (ND)	Sézary syndrome	[84]
*S2454F*	IP3R1	Missense (ND)	Sézary syndrome	[84]
*S2508L*	IP3R1	Missense (ND)	Sézary syndrome	[84]
*G2498S*	IP3R2	Missense: dysfunctional channel *	Anhidrosis	[78]
*R64H*	IP3R3	Missense (ND)	HNSCC	[80]
*R149L*	IP3R3	Missense (ND)	HNSCC	[80]

HNSCC: Head and neck squamous cell carcinoma; ND: Not determined; SCA: Spinocerebellar ataxia; ***** predicted effect on protein.

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
