# Peer review of "Inositol 1,4,5-Trisphosphate Receptors in Human Disease: A Comprehensive Update"

_jcm, 2020, doi:10.3390/jcm9041096_

Round 1

Reviewer 1 Report

In the manuscript (ID#: jcm-755107), titled “ Inositol 1,4,5-trisphosphate Receptors in Human Disease: A Comprehensive Update”, authors, Gambardella et al, reviewed the result from epidemic studies on the association of  ITPR-gene mutations with human diseases.  They concluded that current investigation linked ITPRs to several human diseases; pointed out that several important area regarding ITPRs needs to be further explored. This is a nice review, summarizing the ITPRs’ protein structures and their possible function, will be very interesting to readers. However, I have few suggestions, which are listed in the following paragraphs:

  1. It would be more clear to provide a table to summarize the ITPR mutation location, gain/loss-of-function, linked specific diseases, references. The conclusion made from this summarizing table could benefit more to the readers and other researchers.
  2. ITPRs have three isoforms with similar sequence homology and same domains. Is there any difference in their tissue-distribution? This information may help to understand the specific role of each isoform in in the pathophysiology.
  3. It would be more interesting to provide some information regarding the phenotypes of ITPR-gene knockout mice, to be combined with data from humane GWAS.

Author Response

In the manuscript (ID#: jcm-755107), titled “ Inositol 1,4,5-trisphosphate Receptors in Human Disease: A Comprehensive Update”, authors, Gambardella et al, reviewed the result from epidemic studies on the association of  ITPR-gene mutations with human diseases.  They concluded that current investigation linked ITPRs to several human diseases; pointed out that several important area regarding ITPRs needs to be further explored. This is a nice review, summarizing the ITPRs’ protein structures and their possible function, will be very interesting to readers. However, I have few suggestions, which are listed in the following paragraphs:

  1. It would be more clear to provide a table to summarize the ITPR mutation location, gain/loss-of-function, linked specific diseases, references. The conclusion made from this summarizing table could benefit more to the readers and other researchers. Thanks for your constructive comments; the table (Table 2) has been added, as requested.
  2. ITPRs have three isoforms with similar sequence homology and same domains. Is there any difference in their tissue-distribution? This information may help to understand the specific role of each isoform in in the pathophysiology. This information has been added in Table 1.
  3. It would be more interesting to provide some information regarding the phenotypes of ITPR-gene knockout mice, to be combined with data from humane GWAS. This information is included in Table 2.

Reviewer 2 Report

In their paper Gambardella J et al. examined the role of Inositol 1,4,5-trisphosphate receptors (ITPRs) in the pathogenesis of neurological, immunological, cardiovascular, and neoplastic human diseases.

In the first part, ITPRs structure is well described and disease-related mutations shown in a figure.

In the second part, the authors describe neurological disorders associated with mutations of ITPRs: Spinocerebellar ataxia, Huntington’s Disease and Alzheimer’s Disease, Gillespie syndrome, Autism spectrum disorder and amyotrophic lateral sclerosis and Amyotrophic lateral sclerosis.

The third-fifth paragraphs, the role of ITPRs in autoimmune diseases, anhidrosis and cancer is addressed.

Finally, as described in the sixth paragraph, the genome whole association studies also reports a potential role of this channel in other conditions, like rheumatoid arthritis, type 1 diabetes mellitus, Graves’ disease, cardiovascular disease and allergic disorders including asthma, allergic rhinitis, atopic dermatitis, and airflow obstruction.

The paper is well structured and written. I would suggest to describe some data in Tables.

Author Response

In their paper Gambardella J et al. examined the role of Inositol 1,4,5-trisphosphate receptors (ITPRs) in the pathogenesis of neurological, immunological, cardiovascular, and neoplastic human diseases.

In the first part, ITPRs structure is well described and disease-related mutations shown in a figure.

In the second part, the authors describe neurological disorders associated with mutations of ITPRs: Spinocerebellar ataxia, Huntington’s Disease and Alzheimer’s Disease, Gillespie syndrome, Autism spectrum disorder and amyotrophic lateral sclerosis and Amyotrophic lateral sclerosis.

The third-fifth paragraphs, the role of ITPRs in autoimmune diseases, anhidrosis and cancer is addressed.

Finally, as described in the sixth paragraph, the genome whole association studies also reports a potential role of this channel in other conditions, like rheumatoid arthritis, type 1 diabetes mellitus, Graves’ disease, cardiovascular disease and allergic disorders including asthma, allergic rhinitis, atopic dermatitis, and airflow obstruction.

The paper is well structured and written. I would suggest to describe some data in Tables. Thanks. Two tables have been added.

Reviewer 3 Report

This review paper entitled “Inositol 1,4,5-trisphosphoate Receptors in Human Disease: A Comprehensive Update” tries to summarize the roles of all of three subtypes of ITPRs for human disease partially based on genetic analyses. Though major pathological functions relating to ITPRs are covered in this manuscript presented mainly by the facts with research with human samples, there are several concerns about the manuscript as described below,

Minor comments

  1. On the line 227 of page 6, though the authors mentioned about the role of ITPRs in the cardiovascular fields, the contents described from line 229 to 243 seem to be insufficient. Actually the results of roles of ITPRs for CVDs from genetic analyses have been limited so far, but there are some previous reports related to ITPRs using human patient samples with cardiovascular diseases. In addition, the studies with any subtypes of ITPRs knockout mice or with IP3-sponge transgenic mice exhibited cardiac phenotype in the stage from embryos to adults (reviews, Garcia MI and Boehning D. BBA 1864: 907-914, 2017, Santulli G, et al. J Physiol 595: 3041-3051, 2017). Especially focused on the roles of ITPRs for cardiogenesis, because the abnormalities in the cardiac development lead to lethality in utero, it may be difficult to detect mutations causing severe heart defects by the genetic analyses using patient samples postnatally. Please explain the roles of ITPRs for very early embryogenesis by experiments using neutralizing antibodies, knockout mice and IP3-sponge including the fertilization (Miyazaki S, et al. Science 1992; 257(5067):251-5., DV axis (Saneyoshi T, et al. Nature 2002; 417(6886):295-9) and organogenesis (cardiogenesis) (Uchida K, et al. PLoS One 2010; 5(9)pii:e12500, Nakazawa M, et al. JMCC 2011:51(1):58-66, Uchida K, et al. Dev Biol 2016; 418(1):89-97).
  2. On the line 33 to 34 of page 1, the five domains of ITPR were not described in the ref 9. If you refer to these five domains please cite the appropriate paper.
  3. On the line 72 of page 2, ref 21 did not describe V494I mutation in SCA in Australian family. please correct the citation instead of ref21.
  4. On the line 141 of page 4, though the title of 2.4 is “Autism spectrum disorder and amyotrophic lateral sclerosis, the contents concerning ALS was not explained in this section 2.4. “and amyotrophic lateral sclerosis” should be deleted in the title of 2.4.
  5. On the line 143 of page 4, is ref52 correct? This report analyzed large copy-number variants associated with intellectual disability and congenital abnormalities including autism.
  6. On the line 152 of page 4, please delete the underline of “motor neurons”.
  7. On the line 179 of page 5, please substitute ITPR for IPTR.
  8. On the line 241 to 243 of page 6, the sentence “Finally, in the Hispanic population, ITPR1 was associated to the pathophysiology of childhood obesity [82], while ITPR3 was linked to the body mass index variants conferring a high risk of extreme obesity [80]” does not concern the cardiovascular field. It would be better if the section is divided into two paragraphs.

Author Response

Reviewer 3

This review paper entitled “Inositol 1,4,5-trisphosphoate Receptors in Human Disease: A Comprehensive Update” tries to summarize the roles of all of three subtypes of ITPRs for human disease partially based on genetic analyses. Though major pathological functions relating to ITPRs are covered in this manuscript presented mainly by the facts with research with human samples, there are several concerns about the manuscript as described below,

Minor comments

  1. On the line 227 of page 6, though the authors mentioned about the role of ITPRs in the cardiovascular fields, the contents described from line 229 to 243 seem to be insufficient. Actually the results of roles of ITPRs for CVDs from genetic analyses have been limited so far, but there . In addition, the studies with any subtypes of ITPRs knockout mice or with IP3-sponge transgenic mice exhibited cardiac phenotype in the stage from embryos to adults (reviews, Garcia MI and Boehning D. BBA 1864: 907-914, 2017, Santulli G, et al. J Physiol 595: 3041-3051, 2017). Especially focused on the roles of ITPRs for cardiogenesis, because the abnormalities in the cardiac development lead to lethality in utero, it may be difficult to detect mutations causing severe heart defects by the genetic analyses using patient samples postnatally. Please explain the roles of ITPRs for very early embryogenesis by experiments using neutralizing antibodies, knockout mice and IP3-sponge including the fertilization (Miyazaki S, et al. Science 1992; 257(5067):251-5., DV axis (Saneyoshi T, et al. Nature 2002; 417(6886):295-9) and organogenesis (cardiogenesis) (Uchida K, et al. PLoS One 2010; 5(9)pii:e12500, Nakazawa M, et al. JMCC 2011:51(1):58-66, Uchida K, et al. Dev Biol 2016; 418(1):89-97). WE MODIFIED THE PARAGRAPH AND CITED THE SUGGETED PAPERS.
  2. On the line 33 to 34 of page 1, the five domains of ITPR were not described in the ref 9. If you refer to these five domains please cite the appropriate paper. CORRECTED; THANKS.
  3. On the line 72 of page 2, ref 21 did not describe V494I mutation in SCA in Australian family. please correct the citation instead of ref21. CORRECTED; THANKS.
  4. On the line 141 of page 4, though the title of 2.4 is “Autism spectrum disorder and amyotrophic lateral sclerosis, the contents concerning ALS was not explained in this section 2.4. “and amyotrophic lateral sclerosis” should be deleted in the title of 2.4. DONE; THANKS.
  5. On the line 143 of page 4, is ref52 correct? This report analyzed large copy-number variants associated with intellectual disability and congenital abnormalities including autism. WE REMOVED THAT REF.
  6. On the line 152 of page 4, please delete the underline of “motor neurons”.
  7. On the line 179 of page 5, please substitute ITPR for IPTR.
  8. On the line 241 to 243 of page 6, the sentence “Finally, in the Hispanic population, ITPR1 was associated to the pathophysiology of childhood obesity [82], while ITPR3 was linked to the body mass index variants conferring a high risk of extreme obesity [80]” does not concern the cardiovascular field. It would be better if the section is divided into two paragraphs.